# A mixed-method study of feasibility and acceptability of the Dapivirine vaginal ring among adolescent girls and young women (AGYW) in selected Zimbabwean districts

Noah Taruberekera[1], Malvern Munjoma [1]*, Owen Mugurungi[2], Getrude Ncube[2], Miriam Mutseta[2], Munyaradzi Dhodho[1], Hanul Choi [1], Jabulani Mavudze [1], Tafara Moga[1], Blessing Mutede [1]

**1** Population Solutions for Health, Harare, Zimbabwe, **2** Ministry of Health and Child Care, Harare, Zimbabwe

* malmunjoma@yahoo.com

## Abstract

The Dapivirine vaginal ring (DPV-VR) is an intravaginal silicone ring that delivers an antiretroviral drug (Dapivirine) directly to vaginal tissues for 28 days. This ring protects women against HIV during the receptive vaginal sex at the site of potential infection. In 2021, the WHO recommended DPV as an additional prevention method for high-risk women for HIV with other approaches. With its discreet usage and disposal, DPV-VR has become a preferred HIV method among young women in Sub-Saharan Africa with a prevalent patriarchal social structure that prevents women from making decisions on their bodily autonomy. This study is aimed to assess the acceptability and feasibility of introducing DPV-VR as an HIV prevention method among young women in Zimbabwe and assess motivations and barriers of DPV-VR uptake among target population. We conducted an open-label prospective cohort study from 26 April 2022 to 23 January 2023 across 8 districts in Zimbabwe. Sexually active HIV-negative women aged between 18 and 25 years who were identified as high risk were offered a choice of oral PrEP or DPV-VR. Participants who chose DPV-VR were followed up for six months to receive monthly ring replacement and measure feasibility and acceptability. In-depth interviews were conducted with recipients of care who discontinued, continued for six months, or seroconverted while enrolled in the study to understand their experiences. A minimum of five clients who seroconverted were interviewed to identify potential timeline of seroconversions and risky behaviors. A total of 1,596 eligible participants were enrolled to study, and 1206 (76%) received DPV-VR and 390 AGYW opted for oral PrEP. Continuation rates were comparable among two groups at one month at 83% in the DPV ring arm and 84% in the oral PrEP arm. At 6 months, 64% of DPV users continued, compared to 16% in the oral PrEP arm. Participants who preferred to self-insert the ring increased from 50% at one-month follow up to 85.4% at 6 months. Seroconversion rates were comparable across two groups, as 9 out of 1095 (0.82%) DPV-VR users were seroconverted compared to 2 out of 390 (0.51%) oral PrEP users (p=0.608). Some DPV users mentioned pelvic pain and lower abdominal pain as common side effects. In in-depth interviews, participants mentioned motivators for

**Data availability statement:** The data is available upon contacting the corresponding authors. The datasets are accessible on the DRYAD on the following link: https://datadryad.org/stash/share/ywv_6d3e_rRH7H3A-aNNuW-tyVAEPpPNyvlGXhkgG9CU

**Funding:** This study was financially supported by the United States Agency for International Development (USAID) [https://www.usaid.gov] in the form of a grant (72061323CA00002) awarded to BM. This study was also financially supported by Population Solutions for Health (PSH) [https://psh.org.zw] in the form of salaries for NT, MM, MD, HC, JM, TM. The specific roles of these authors are articulated in the 'author contributions' section. The funders had no role in study design, data collection and analysis, decision to publish, or preparation of the manuscript.

**Competing interests:** The authors have read the journal's policy and have the following competing interests: NT, MM, MD, HC, JM, TM are employees of Population Solutions for Health (PSH) [https://psh.org.zw]. This does not alter our adherence to PLOS policies on sharing data and materials. There are no patents, products in development or marketed products associated with this research to declare.

DPV uptake such as its discreet use and not having to take medication daily. They also recommended to develop rings that last longer than current 28-day lifespan for women in rural areas or mobile who do not have continuous access to resources. This research provides evidence of DPV-VR as an acceptable and feasible HIV prevention in LMICs. Clients found it easy to insert the ring by themselves, and it provides a discreet way to protect themselves from HIV infection. Nevertheless, there are social barriers that hinder women's decision-making power in protecting their bodies. Therefore, it is recommended to conduct further studies to identify solutions for barriers and scale-up.

## Introduction

HIV burden remains high in Zimbabwe. Most recent nationally representative data show the prevalence of 12.9% among individuals aged 15 to 64; 10.2% among males and 15.3% among females [1]. Only 86.8% of people living with HIV (PLHIV) aged 15 to 64 years report knowing their HIV status, and the annual HIV incidence is 31,000 new cases of HIV among Zimbabweans [1]. Adolescent girls are disproportionately affected with an HIV incidence of 0.54% compared to 0.13% of their male counterparts. Therefore, it is important to scale up combination prevention options for adolescents and young women.

The DPV-VR is an intravaginal silicone ring developed by the International Partnership for Microbicides (IPM) for HIV prevention [2]. The ring delivers an antiretroviral drug called Dapivirine that is continuously released for 28 days directly to vaginal tissue, which helps to protect against HIV at the site of potential infection, only under exposure from receptive vaginal sex. As part of the revised WHO Consolidated HIV Guidelines for Prevention, Treatment, Service Delivery & Monitoring released in July 2021, the WHO recommends that DPV-VR may be offered as an additional prevention choice for women at substantial risk of HIV infection as part of combination prevention approaches [3]. This user-initiated method could be an alternative solution for those who cannot or do not want to take oral PrEP. Efficacy studies show that Dapivirine reduces the risk of HIV infection by 35% [2]. The ring is currently approved for women 18 years and older since the younger adolescents were not included in the original trials, and the evidence on safety in pregnant and breastfeeding women is sparse yet reassuring. Among 169 participants who became pregnant during the original trial, no congenital, maternal, or infant adverse outcomes were observed [2].

DPV-VR is increasingly becoming a preferred HIV prevention method. There were two studies conducted in Kenya, South Africa, and Zimbabwe (TRIO and Quatro, respectively) on the optimal way to deliver PrEP for women among placebo, vaginal rings, oral PrEP and long-lasting injectable ARV agents [4,5]. The TRIO study found that while injectables were most preferred, continuation rates and willingness of uptake were higher among ring participants [4]. Similarly, the Quatro study concluded that initial preference for the ring was low but increased with trial duration and had a high adherence rate. This preference trend varied by population and combination approach, so it is necessary to inform advantages of DPV-VR to specific populations to better inform the scaling up of this recent intervention. In a double-blind placebo-controlled clinical trial conducted across 15 sites in Malawi, South Africa, Uganda, and Zimbabwe, researchers identified an association between the acceptability of the method and subsequent adherence [6]. They determined adherence by residual Dapivirine in the returned vaginal rings and identified factors associated with non-adherence such as effects on sex, perceived negative change to the vaginal environment, and concerns about wearing the ring during menstruation [6]. In a double-blind placebo-controlled clinical trial conducted across 15 sites in Malawi, South Africa, Uganda, and Zimbabwe, researchers identified

an association between the acceptability of the method and subsequent adherence [6]. They determined adherence by residual dapivirine in the returned vaginal rings and identified factors associated with adherence such as effects on sex, perceived negative change to the vaginal environment, and concerns about wearing the ring during menstruation [6].

The role of the primary sexual partner in deciding to use DPV-VR was also investigated by several studies [7,8]. In low- and middle-income countries (LMICs) like Zimbabwe, patriarchal social structure and gender norms position men's influence over women in decision-making for HIV prevention [7]. Thus, understanding the influence of the male partners in DPV-VR is critical among high-risk women with regular sexual partners. Furthermore, other social and structural factors, such as the desire to have a child, experience of social harm or intimate partner violence, initial worries about a novel prevention method, and changed perception of HIV risk, may also play a role in the acceptability of DPV-VR [8]. Also, it is important to understand individuals' preferences on product design dosing regimens and ways to deliver the additional dose to increase uptake and reduce stigma. Therefore, this study is aimed to assess the acceptability and feasibility of introducing DPV-VR as an HIV prevention method among young women in a resource-limited setting such as Zimbabwe and assess motivations and barriers of DPV-VR uptake to create approaches for the service delivery.

## Methodology

This study was an open-label prospective cohort study conducted from 26 April 2022 to 23 January 2023. HIV-negative individuals screened as high risk were offered a choice of oral PrEP or DPV-VR. Reasons for choosing a method were documented, but only those who chose the DPV-VR were followed for six months to measure continuity on the method and user experiences for feasibility and acceptability. Rings were delivered using the current PrEP delivery (facility and home-based) platforms to assess approaches that work for the target population. Fig 1 below shows the study recruitment flow chart.

### Procedures

In each district, we used HIV screening assessment tools to identify potential study participants. The initial sampling frame consisted of AGYW accessing HIV and or sexual and reproductive health services public sector health facilities and Population Solutions for Health funded health care centres (New Start Centres). The recruitment strategy was a random sample of participants voluntarily consenting to receive either oral or DPV-VR as their preferred PrEP method. Individuals who consented to receive DPV-VR for at least one month and consented to participate in the study were enrolled. Participants received instructions on the ring and were given a choice of provider-delivered or self-insertion. Face to face structured interviews were conducted and participants were requested to return to health facilities a month later to insert another ring or collect a ring for self-insertion and a follow-up interview. Additionally, in-depth interviews were conducted with lapsed users, consistent users, and those who decided to drop out at earlier stages to understand their experiences. Five clients who seroconverted after participating in the study were interviewed to get an in-depth understanding of the circumstances surrounding the seroconversions.

### Outcome measures

Acceptability was measured by the proportion of high-risk women voluntarily choosing DPV-VR after being offered the oral PrEP option and feasibility by HIV incidence per 100 person years. Qualitative outcomes focused on participants' experiences with using DPV-VR from time of insertion to continuation, lapsing and or discontinuation.

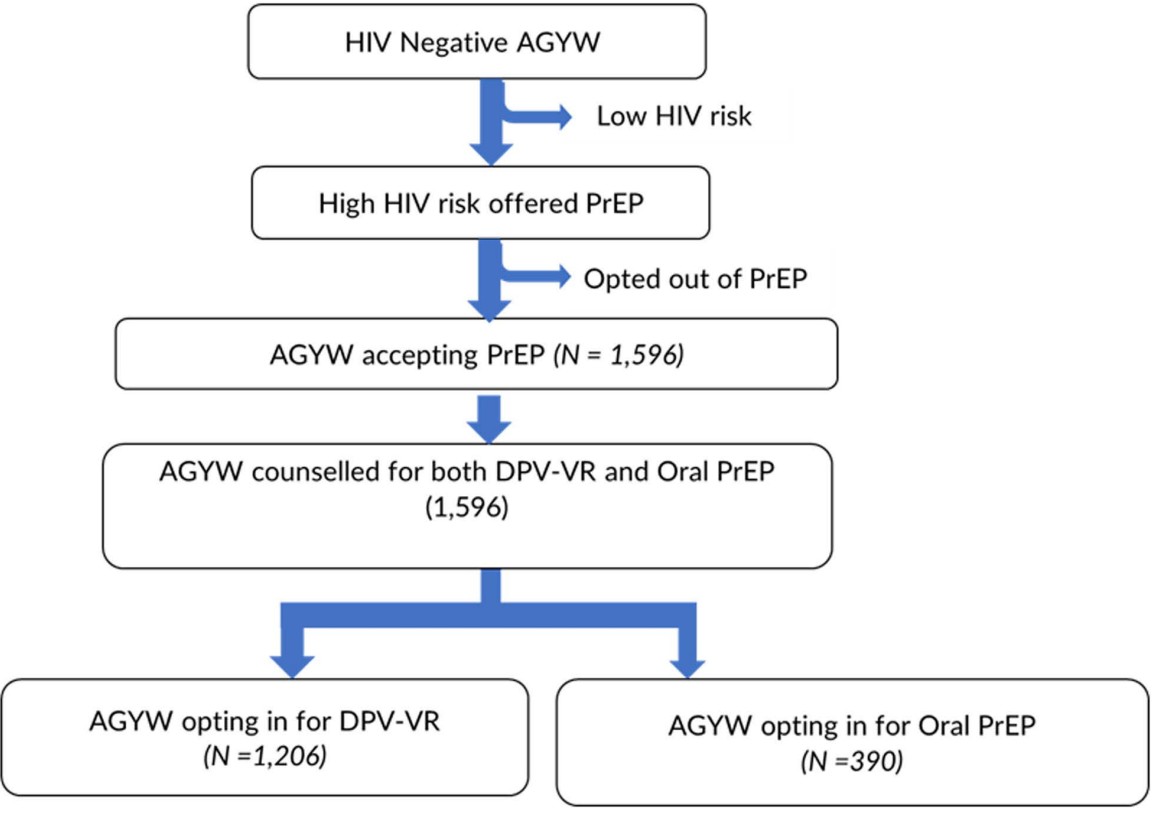

**Fig 1.  Client screening and recruitment data flow.**

### Ethical considerations

This study received ethical approval from the Medical Research Council of Zimbabwe (MRCZ/A/2846). No deviation from the protocol was implemented without prior review and approval of the institutional review board (IRB) except where it was necessary to eliminate an immediate hazard to a research subject. In such a case, the deviation would be immediately reported to the IRB. Nonetheless, no such incidents nor adverse events were reported in the duration of the study. Participants received compensation of US$10 for their time spent on this study. Written informed consent was obtained from all study participants, and participants were provided a copy of their informed consent form.

### Confidentiality

Participants' information was managed with respect and confidentiality during all research procedures and publications. Only de-identified data and interview transcripts were used for analysis and reviewed by researchers prior to publication to ensure that the risk of breach of confidentiality was managed and minimal. After completion of study, information will be kept securely at the project office for 12 months after completing all analyses.

## Results

### Quantitative results

1,596 eligible participants were enrolled in the study, and 1,206 (76%) opted for DPV-VR and 390 AGYW opted for oral PrEP. Table 1 shows that almost 40% of participants were aged

between 18 and 20 for both DPV and oral PrEP groups, and the rest of the age bands were also similar across the two study arms. The cumulative proportion of participants who had completed at least primary school was comparable across the two groups. Proportions of individuals who are married or cohabiting were nearly a third for both DPV and oral PrEP users. Significantly higher proportions of individuals in rural areas chose DPV (73.5%) compared to 36.9% among oral PrEP users. Religion was similar in both arms, except the apostolic sect which was 5.9% among DPV -VR users and 3.3% among the oral PrEP users. The oral PrEP arm had a higher proportion (29.4%) of respondents reporting at least 5 sexual partners in the past 6 months compared to just 9.6% in the DPV-VR arm.

**DPV ring acceptability at enrollment.** Table 2 shows that Chipinge district (predominantly rural) had the highest DPV-VR acceptability among participants at enrollment (91%, 95% CI: 87.6 - 93.5), and Bulawayo (urban district) had the lowest DPV VR acceptance rate at 52% (95% CI: 46.9-57.4), followed by Gweru as shown in the table below.

**PrEP continuation rates among DPV and oral PrEP users.** Continuation rates were comparable among two groups at one month at 83% (95% CI 80.3-84.6) in the DPV-VR arm and 84% (95% CI 80.1-87.6) in the oral PrEP arm as shown in Fig 2 below. At 3-month follow-up, continuation rates among oral PrEP users were 27% lower than DPV-VR users. At 6 months, 64% (95% CI: 60.6-66.1) of DPV users continued, compared to 16% (95% CI: 12.6-20.2) in the oral PrEP arm.

**Self-insertions over time.** The proportion of participants who preferred self-insertion increased from 50% (95% CI: 46.6-52.9) at one-month follow up to 85.4% (95% CI: 82.7-87.8) at 6-month follow-up as presented in Fig 3 below.

**HIV sero conversions and HIV incidence.** HIV sero-conversions and HIV incidence are presented in Table 3 below. Nine out of 1095 1095 (0.82%, 95% CI: (0.42-1.57)) DPV-VR users were seroconverted compared to 2 out of 390 (0.51%, 95% CI: 0.06 – 1.84) oral PrEP users.

**Table 1. Demographic profile of study participants.**

| Variable | Category | DPV users | | Oral PrEP users | |
|---|---|---|---|---|---|
| | | Frequency | Proportion (%) | Frequency | Proportion (%) |
| **Age** | 18 – 20 | 461 | 38.2 | 154 | 39.5 |
| | 21 – 22 | 359 | 29.8 | 119 | 30.5 |
| | 23 – 25 | 386 | 32.0 | 117 | 30.0 |
| **Level of Education** | None | 17 | 1.4 | 0 | |
| | Primary School | 315 | 26.1 | 45 | 11.5 |
| | Secondary School | 857 | 71.1 | 316 | 81.0 |
| | Vocational/University | 17 | 1.4 | 29 | 7.4 |
| **Marital Status** | Married/ Cohabiting | 347 | 28.8 | 104 | 26.7 |
| | Separated/ Divorced | 145 | 12 | 24 | 6.2 |
| | Widowed | 3 | 0.3 | 0 | 0.0 |
| | Single/in a relationship | 711 | 52.0 | 262 | 67.2 |
| **Setting** | Rural | 886 | 73.5 | 144 | 36.9 |
| | Urban | 320 | 26.5 | 246 | 63.1 |
| **No. of sex partners in the last 6 months** | 0 | 1 | 0.1 | 0 | 0.0 |
| | 1 | 468 | 38.8 | 23 | 5.9 |
| | 2-3 | 535 | 44.4 | 199 | 51.0 |
| | 4 | 86 | 7.1 | 54 | 13.7 |
| | 5 or more | 116 | 9.6 | 115 | 29.4 |
| **Total** | 1206 | 1209 | 100 | 390 | 100 |

**Table 2. DPV-VR acceptability relative to Oral PrEP.**

| Location | Setting | Total PrEP (N) | Total DPV-VR (N) | Proportion of DPV-VR compared to PrEP participants (%) | 95% CIs (%) |
|---|---|---|---|---|---|
| Bulawayo | Urban | 364 | 190 | 52% | 46.9 - 57.4 |
| Gweru | Urban | 214 | 130 | 61% | 53.9 - 67.3 |
| Chipinge | Rural | 405 | 368 | 91% | 87.6 - 93.5 |
| Mutare | Rural + Urban | 299 | 257 | 86% | 81.5 - 89.7 |
| Mat South | Rural | 314 | 261 | 83% | 78.5 - 87.1 |
| **Total** | | **1596** | **1206** | **76%** | **73.4 -77.6** |

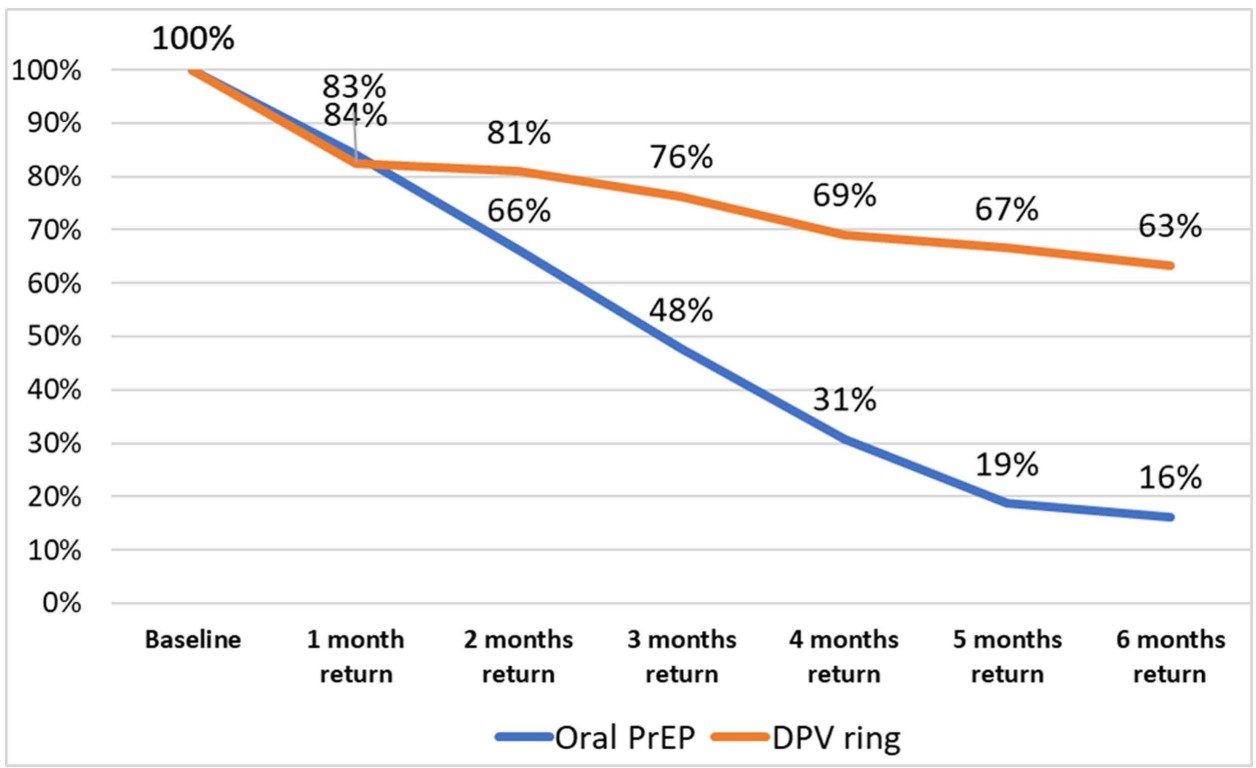

**Fig 2. PrEP continuation rates among DPV-VR and oral PrEP users.**

HIV incidence was also comparable between the two groups, 1.79 (95% CI: 0.93-3.43) per 100 person-years among DPV-VR users compared to 0.67 (0.17-2.69) per 100 person-years among oral PrEP users.

Within DPV users, six seroconversions occurred at one-month follow-up and other three sero conversions at two, three, and four-month follow-ups. Five clients who seroconverted in the first month all reported that they did not remove the ring at any point. Individuals were in a similar age group, and seven lived in the rural area.

Table 4 profiles HIV sero-conversions by follow-up period, age and residence. Among DPV-VR users, six seroconversions occurred at one-month follow-up and other three sero-conversions at two, three, and four-month follow-ups each. Five clients who seroconverted in the first month all reported not removing the ring at any point. Five clients were aged 20-25

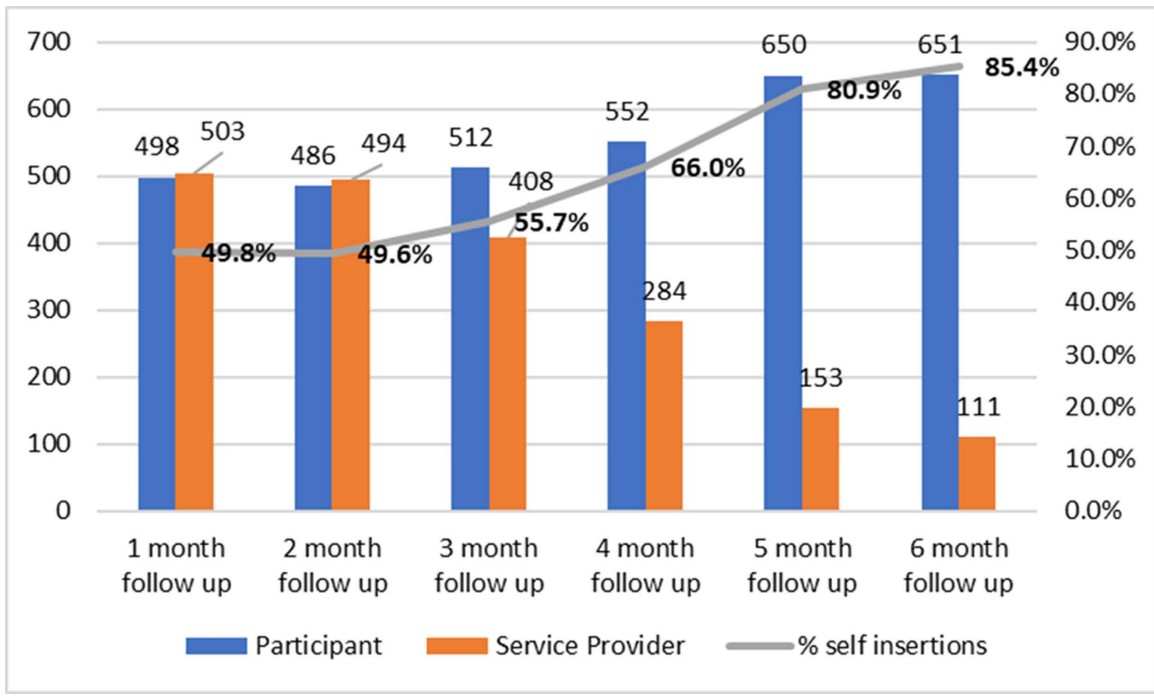

**Fig 3. Self insertions trend over the six months follow-up period.**

**Table 3. HIV sero conversions between DPV-VR and Oral PrEP users.**

| Method | Total users | Number of participants sero-converted | % sero-converted (95% CI) | Incidence rate per 100 person years (95% CI) |
|---|---|---|---|---|
| DPV-VR | 1,180 | 9 | 0.76 (0.35 - 1.44) | 1.79 (0.93-3.43) |
| Oral PrEP | 390 | 2 | 0.51(0.06 – 1.84) | 0.67 (0.17-2.69) |

and the other four were below 20 years of age. There was no sufficient evidence to detect any statistical differences across the presented demographics for seroconversion.

**Side effects among DPV users.** Fig 4 presents frequencies of side effects reported by participants using DPV-VR. The most reported side effects among DPV users were pelvic pain and lower abdominal pain, and other side effects included headaches, period pain, pain during sex and UTI.

## Qualitative results

Participants who were enrolled for the qualitative interviews were either current users of the ring and never defaulted, lapsed users, or seroconverted while using the DPV-VR. In-depth interviews were conducted with 14 participants with 6 initially recruited but discontinued before 6 months, 5 enrolled and continued with the DPV-VR for 6 months and did not miss appointments, and 3 received DPV-VR initially but seroconverted during the follow-up. All the participants were not married, three had at least one child, and ten had none.

**Participants who continued DPV-VR.** Participants who continued on DPV-VR confirmed that it had several advantages over oral PrEP. All four respondents found it easy to use, comfortable, and less likely to interfere with systemic medicines. Also, continuous use of

**Table 4. Profile of HIV sero conversions among DPV users.**

| Variable | Categories | HIV sero-conversions | Total followed up | % (95% CIs) |
|---|---|---|---|---|
| **Follow-up period** | One month | 6 | 1030 | 0.58(0.26 - 1.29) |
| | Two months | 1 | 992 | 0.10 (0.01 – 0.71) |
| | Three months | 1 | 926 | 0.11(0.02 - 0.76) |
| | Four months | 1 | 841 | 0.12(0.02-0.84) |
| **Age** | 18-20 | 5 | 417 | 1.2(0.50-2.85) |
| | 21-22 | 0 | 328 | – |
| | 23-25 | 4 | 350 | 1.14(0.43-3.0) |
| **Residence** | Rural | 7 | 828 | 0.85(0.40-1.76) |
| | Urban | 2 | 267 | 0.75(0.19-2.95) |

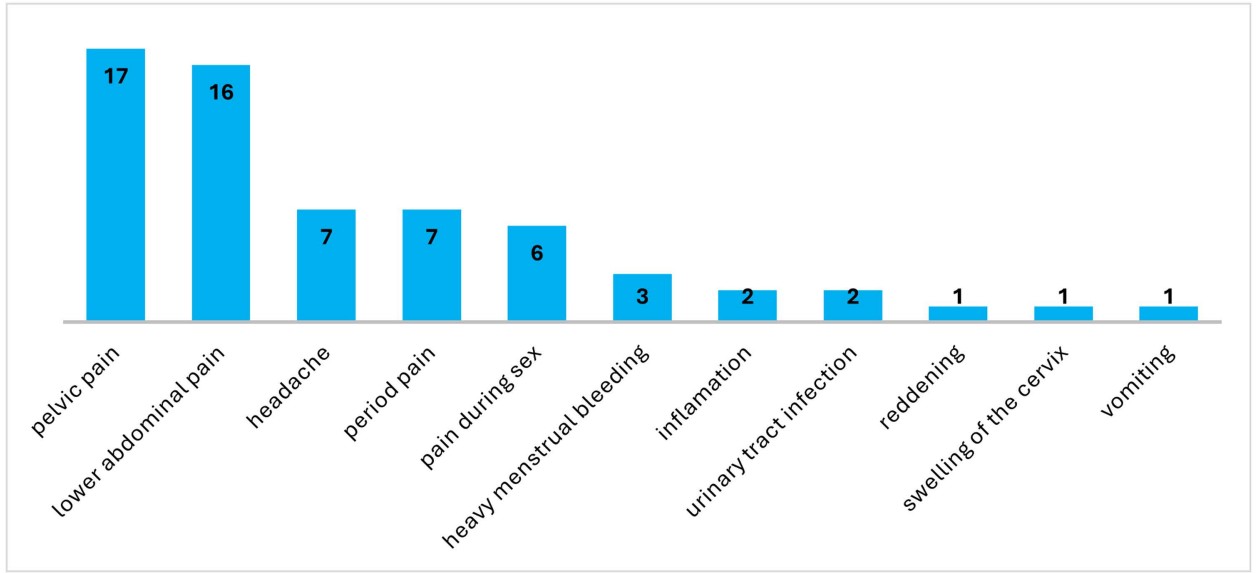

**Fig 4. Side effects among DPV users.**

the DPV-VR was reported to be easy as it was taken just once a month unlike oral PrEP which is a daily pill.

Another respondent mentioned fears of sexual abuse and added that there was no need to go to the clinic to receive post-exposure prophylaxis (PEP) with the DPV-VR. One of the motivators for use was that it was a discrete method and AGYW still under the guardian of conservative parents appreciated this convenience. Young women with low self-efficacy to condom negotiation felt empowered. One young woman from Chipinge said,

> *"You know young women like us are having more sexual activities than the older married women, yet we still go to school, so the ring is much more secretive, I can be in class learning and no one knows I am wearing it, unlike pills that are risky because someone will catch you one day (AGYW, Chipinge).*

To increase and maintain uptake, participants suggested giving multi-month supplies of the DPV-VR and more research to develop rings that can last longer than the current 28-day

lifespan. They recommended hosting road shows in urban areas that could potentially attract more users, underscoring its fewer side effects and effectiveness in HIV prevention. Another suggestion was hosting school-level campaigns targeting women and girls than hosting random community campaigns. WhatsApp was most often mentioned as a mode of promotion.

**Participants who discontinued DPV-VR.** Six participants who discontinued on the DPV-VR mentioned that traveling and other commitments made them unable to come for follow-up visits. Similar to the previous group, discontinued participants pointed out that adherence could be improved by having a longer lasting ring or receiving multi-month dispensing (MMD) of the 28-day rings. A young woman from Mutare mentioned this limitation.

*"I travelled to Zambia shortly after insertion. I was so confused when my 28 days were up because I wanted to continue to be sexually active in Zambia and I did not like condoms anymore" (Mutare, AGYW).*

Some participants indicated DPV-VR's inability to protect them from STIs as a major limitation of the method. In addition to the non-protective effect of the ring against STIs, two respondents experienced pain when they inserted the ring, and this prompted them to remove it before 28 days. To improve uptake of the ring, they recommended to distribute among DREAMS recipients and provide discreet, private places to insert rings.

**Participants who seroconverted while on DPV ring.** Three participants who seroconverted while using the ring were interviewed. One participant tested positive when she was about to get her second insertion, and she mentioned she might have been already positive at enrollment. Since she was a sex worker, she could not suspect anyone to have transmitted the virus to her and acknowledged that she was at an elevated risk of getting infected. Another HIV-positive RoC mentioned that she removed the ring when her partner was returning home from abroad, due to fear that her partner might feel the ring and could be violent towards her if he found out she was on the ring. The participant had this to say;

*"My boyfriend is based in South Africa; he was coming home for the holidays and I removed the ring because I knew the story would not end well. He is very violent. I regret this choice because I think that was when he gave me HIV" (Bulawayo, 23 year old).*

The last seroconverted participant stated that she believed the DPV-VR made the monthly period longer to seven days instead of the usual three to four days. She tested HIV positive on the fourth visit and was skeptical about the effect of the DPV-VR on its full protection against HIV. All three participants confirmed that it provided them privacy and could be a preferred PrEP method for the public.

## Discussion

We conducted an open-label prospective cohort study from June 2022 to June 2023 to evaluate acceptability and feasibility of young women in implementing DPV-VR as a new HIV prevention method in a low-resource setting like Zimbabwe. This study showed comparable HIV positivity and seroconversions between DPV-VR and oral PrEP cohorts. We observed high acceptability rates among the rural participants compared to their urban counterparts. Furthermore, participants in the DPV-VR cohort had higher continuation rates compared to the oral PrEP cohort at the six-month follow-up.

There were nine seroconversions in the ring cohort, with six observed in the first month. It is highly likely that these sero-conversions occurred before DPV-VR insertion as the participants were in HIV infection window period at the time of enrollment. Sero-conversion did

not differ by age and residence since the rural sub-sample with eight sero conversions, had a proportionally larger sample size. The total number of sero-conversions was not large enough to test the difference between rural and urban respondents on seroconversion likelihood.

Two participants mentioned removing the ring and having unprotected sex because they did not want their partner to find out they were using the ring. This barrier is corroborated in a study from South Africa where fear of discovery by partner was a barrier to continuous ring use [9,10]. Addressing intimate partner violence and gender-based power imbalances is essential for the DPV ring's acceptability. Online resources promoting healthy relationships can help mitigate partner violence, facilitating greater adoption and acceptance of the DPV-VR [9,10].

A comparable DPV acceptability rate and HIV positivity were observed in other studies. For instance, a study in South Africa, Zimbabwe, and Uganda showed 67% acceptability among DPV users [11]. Also, HIV incidence results across various demographics were comparable to an open-label extension phase II clinical trial at 14 sites in Malawi, South Africa, Uganda, and Zimbabwe (2.7 per 100 person-years) and phase 3B clinical trial multicentre, follow-on, open-label extension trial (1.8 per 100 person-years) [12,13].

This study highlights significant findings on the acceptability of the DPV-VR by AGYW. Sharma and Hill [14] highlight that long-acting cabotegravir could provide more affordable and sustainable HIV prevention compared to traditional oral PrEP regimens. However, as with the DPV-VR, acceptability is a key consideration, with specific groups showing different levels of adherence based on socio-behavioral factors [15].

The findings suggest that AGYW in rural areas had higher acceptability compared to those in urban areas. In addition, AGYW aged 18-20 had higher acceptability of DPV-VR compared to those aged 21-22 or 23-25. Majority of respondents were Christian – a reflection of the general population. The ring seemed acceptable even among religions known to shun health care services such as the African apostolic sect and this may act as an entry point for promotion of other health services in this religious sect. A study conducted in Malawi reinforced the importance of partner dynamics in influencing the use of DPV-VR, the authors argue that male partner support significantly affects adherence and therefore it is essential to address relationships power dynamics in order to improve uptake and acceptability of DPV-VR [16].

The most common side effects of DPV-VR reported included pelvic pain, lower abdominal pain, headaches, and period pain. Qualitative findings showed that participants who continued using the DPV-VR found it easy to use, comfortable and discrete while those who did not continue mentioned travel, commitments, and pain during insertions. A study by Ismail et al. [16] provides insights into user perspectives on microarray patches for PrEP delivery, which can be compared to the DPV-VR in terms of user-friendliness and comfort. Their research highlights the importance of improving long-acting PrEP methods based on feedback from targeted users, similar to how the DPV-VR has been found to be acceptable among users who found it to be discrete and easy to use [17]. These insights are important considering the barriers to continued use such as discomfort during insertion and side effects.

This research provides evidence of DPV-VR as an acceptable and feasible HIV prevention in LMICs. Clients found it easy to insert the ring by themselves, and it provides a discreet way to protect themselves from HIV infection. Nevertheless, there are social barriers that hinder women's decision-making power in protecting their bodies. Therefore, we suggest conducting further studies to identify solutions for barriers and scale-up.

This investigation had some limitations that should be acknowledged. First, we did not collect used rings to assess residual drug levels, which could have provided a surrogate marker for adherence. Without this data, we are not certain that participants did not remove the ring at some point and engaged in unprotected sex. We therefore relied on reported information

that participants gave at follow up. Investigators agreed that this limitation is not major since a more in-depth understanding of the circumstances surrounding sero-conversions was done qualitatively with all sero-converters.

Second, HIV incidence point estimates were less precise particularly for oral PrEP users due to the smaller sample size and limited person-years of follow-up. This explains an insignificant p-value of 0.608 and overlapping confidence intervals across the two arms. Our study had only 286 person-years of follow-up, which is considerably less than the 600 person-years typically observed in similar studies [13,17]. This reduced statistical power may limit the generalizability of our findings with regards to this secondary outcome.

Additionally, we did not assess the HIV sero-status of participants who discontinued the study early. As a result, we cannot ascertain whether some discontinuers acquired HIV after leaving the study, which may have introduced bias. Further research may be required to answer these secondary objectives with more certainty.

Further, there is a lack of data on providers' attitudes toward DPV-VR. Since providers are key to successful intervention delivery, assessing their attitudes is critical for scaling up DPV as a HIV prevention method. Lastly, this study was conducted only with young women, so the response on acceptability and barriers may not be representative of the general population. Hence, it is recommended to identify barriers among the general population before scale-up.

## Conclusions and recommendations

The ring has demonstrated high acceptability rates, particularly in predominantly rural settings, when compared to urban areas. This suggests that DPV-VR may address unique challenges faced by rural populations in accessing HIV prevention methods and this PrEP modality should be scaled up accordingly.

PrEP continuation has been a major challenge especially among young women in Zimbabwe due to pill burden and fear of involuntary disclosure. The study showed higher continuation rates among DPV-VR users compared to oral PrEP users showing that the ring is an immediate remedy to ensuring continuous protection among the high-risk group. Further, low sero-conversion rates, minimal side effects, and the ease of use all contribute to its potential as a game changer in the fight against HIV among high-risk populations.

Respondents have reported high levels of motivation to use the DPV-VR due to its discreet nature and the absence of a daily pill burden, which can often be a barrier with oral PrEP. Despite this enthusiasm, some initial hesitancy remains, likely due to DPV-VR being a new intervention. Education and sensitization efforts are strongly recommended as this will be crucial to overcoming initial hesitancy in uptake.

In conclusion, we recommend scaling up the DPV-VR to other districts as part of a comprehensive combination HIV prevention approach within the national HIV prevention program. The expansion of DPV-VR can further strengthen the prevention landscape, particularly for AGYW, and improve overall HIV prevention outcomes in areas where other methods have faced challenges.

## Supporting information

**S1 Data. This section describes the datasets submitted as part of the supporting information file labelled "S1 Data".** The following STATA 17 files are saved in this folder. 1. "Baseline DPV ring dataset" – contains baseline data for all participants enrolled in the DPV study at baseline. 2. "Follow-up DPV dataset_notmatched" – contains follow up dataset for all participants enrolled in the DPV study, not matched to the baseline study. 3. "matched_baseline_follow up to upload final" this dataset contains matched baseline and follow up datasets for all

DPV study participant. 4. "Oral PrEP Full dataset" – this dataset contains matched baseline and follow up datasets for all study participants enrolled for Oral PrEP in the study. (ZIP)

## Acknowledgments

We would like to thank the United States Agency for International Development for the funding the Ministry of Health and Child Care (MoHCC) and the National AIDS Council of Zimbabwe for their guidance and technical support. We also thank the data collection team including the whole research staff for their commitment to this study. Finally, heartfelt gratitude to the study participants for their voluntary consent to be part of this research study.

## Author contributions

**Conceptualization:** Noah Taruberekera, Owen Mugurungi, Getrude Ncube.

**Formal analysis:** Malvern Munjoma, Munyaradzi Dhodho, Hanul Choi.

**Funding acquisition:** Blessing Mutede.

**Investigation:** Noah Taruberekera, Malvern Munjoma, Owen Mugurungi, Getrude Ncube, Miriam Mutseta, Tafara Moga.

**Project administration:** Malvern Munjoma, Miriam Mutseta, Jabulani Mavudze, Tafara Moga, Blessing Mutede.

**Supervision:** Noah Taruberekera, Owen Mugurungi, Jabulani Mavudze, Tafara Moga, Blessing Mutede.

**Validation:** Malvern Munjoma.

**Visualization:** Munyaradzi Dhodho, Hanul Choi.

**Writing – original draft:** Malvern Munjoma, Munyaradzi Dhodho, Hanul Choi.

**Writing – review & editing:** Hanul Choi, Jabulani Mavudze.

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
