## [Decision Letter · Decision Letter 0]

15 Aug 2024

PGPH-D-24-01437

A mixed-method study of feasibility and acceptability of the Dapivirine vaginal ring among adolescent girls and young women (AGYW) in Zimbabwe

Dear Dr. Munjoma,

Thank you for submitting your manuscript to PLOS Global Public Health. After careful consideration, we feel that it has merit but does not fully meet PLOS Global Public Health’s publication criteria as it currently stands. Therefore, we invite you to submit a revised version of the manuscript that addresses the points raised during the review process.

We look forward to receiving your revised manuscript.

Kind regards,

Tsitsi B. Masvawure, Ph.D.

Academic Editor

Journal Requirements:

Additional Editor Comments (if provided):

Thank you for your submission. Overall, your paper addresses an important topic in HIV prevention and provides critical insights into biomedical prevention technologies that may be relevant and acceptable to adolescent girls and young women. Below are some areas that need further attending to:

1. Methodology: please explain where the AGYW were recruited from. Did you recruit women attending health facilities? Did you partner with local organizations? The recruitment strategy is not explained in the paper and yet is essential for the interpretation of your results.

2. Table 2 is confusing. Your 'Total PrEP' column adds up to 1,596, which is the grand total of study participants; however, only 390 participants opted for PrEP. Please clarify. Furthermore, shouldn't your DPV acceptability vs PREP acceptability by location be based on the total N for each location? For instance, what proportion of participants from Bulawayo opted for DPV vs PrEP, in which case your total N would be the total number of participants from Bulawayo and so on for the other locations? Anyway, please review Table 2 and explain your decisions or the math that informs your analysis in a footnote for the reader.

3. Qualitative results: One key strength of your paper is your mixed-method design. Could you provide some participant excerpts to support the narrative that you provided. This will help augment the value of qualitative research in your study. In the qualitative section, you refer to participants as 'recipients of care' but do not use this term when reporting your quantitative data. I suggest that you simply use the term 'participants' unless you explain how your participants are 'recipients of care', that is, what is the care that they are receiving outside the context of the study.

4. You report that the HIV incidence among DPV users and PrEP users was 'comparable': it is not clear how you are using the word 'comparable'. Additionally, the HIV incidence among DPV users is THREE TIMES that of PrEP users (2.32 per 100 person years vs 0.67 per 100 person years). This is a hardly 'comparable'; please revise or explain what you mean. Did you carry out any statistical tests to assess if this difference is statistically significant? If you did, please include the p-value for this.

5. Discussion: this section needs to be strengthened. Tell the reader what the implications of your findings are: what do we learn about the factors that contribute to DPV acceptability?

6. Please do a thorough copyedit of the paper. There were several small grammatical errors that distracted from an otherwise engaging paper.

Please also see, and respond to the comments from the reviewer below.

Reviewers' comments:

Reviewer's Responses to Questions

**Comments to the Author**

1. Does this manuscript meet PLOS Global Public Health’s publication criteria ? Is the manuscript technically sound, and do the data support the conclusions? The manuscript must describe methodologically and ethically rigorous research with conclusions that are appropriately drawn based on the data presented.

Reviewer #1: Yes

2. Has the statistical analysis been performed appropriately and rigorously?

Reviewer #1: No

3. Have the authors made all data underlying the findings in their manuscript fully available (please refer to the Data Availability Statement at the start of the manuscript PDF file)?

Reviewer #1: No

4. Is the manuscript presented in an intelligible fashion and written in standard English?

Reviewer #1: Yes

5. Review Comments to the Author

Reviewer #1: The manuscript entitled ‘A mixed-method study of feasibility and acceptability of the Dapivirine vaginal ring among adolescent girls and young women (AGYW) in Zimbabwe’ presents exciting findings and can contribute to knowledge in this important area. However, one notes some key shortcomings which the authors will need to address before I can recommend that it be accepted, these are:

1. The authors failed to recognize that religion is a key variable in health-seeking behaviour and adherence in Zimbabwe, especially for individuals belonging to certain apostolic groups. The authors need to report on the study population's religious group variable and use this lens in their analysis, discussions and recommendations.

2. We also want to know the alcohol and drug use patterns of the sample and this needs to be included in the analysis.

3. The authors failed to outline clearly the limitations of their study, this is a key weakness of the manuscript

4. The title ‘A mixed-method study of feasibility and acceptability of the Dapivirine vaginal ring among adolescent girls and young women (AGYW) in Zimbabwe’ is misleading as it implies that a statistically representative sample covered the whole country. The title needs to be changed to something like ‘A mixed-method study of feasibility and acceptability of the Dapivirine vaginal ring among adolescent girls and young women (AGYW) in selected study site in Zimbabwe

5. The manuscript fails to make any detailed recommendations to policymakers in Zimbabwe and other countries in southern Africa other than stating the obvious.

6. I would like to congratulate the authors for ensuring that Zimbabweans are given key positions in this manuscript, i.e., first, second, and last. It's important for Zimbabweans to lead research being conducted in their own country.

All the best

6. PLOS authors have the option to publish the peer review history of their article (what does this mean? ). If published, this will include your full peer review and any attached files.

**Do you want your identity to be public for this peer review?** For information about this choice, including consent withdrawal, please see our Privacy Policy .

Reviewer #1: **Yes: ** Godfrey N Musuka, DVM, PhD

---

## [Editor Report · Decision Letter 1]

30 Dec 2024

A mixed-method study of feasibility and acceptability of the Dapivirine vaginal ring among adolescent girls and young women (AGYW) in selected Zimbabwean districts

PGPH-D-24-01437R1

Dear Mr Munjoma,

We are pleased to inform you that your manuscript 'A mixed-method study of feasibility and acceptability of the Dapivirine vaginal ring among adolescent girls and young women (AGYW) in selected Zimbabwean districts' has been provisionally accepted for publication in PLOS Global Public Health.

Best regards,

Tsitsi B. Masvawure, Ph.D.

Academic Editor

Dear Authors,

Thank you for your patience. We have reviewed your revisions and believe that you responded to the reviewers sufficiently. There are a couple of minor changes to make. First, you mention that participants were assigned to oral PrEP, DPV ring or condoms. I suggest that you delete the condoms part as you seem to have revised this in some parts of the paper but not in others. If, indeed another group was assigned condom use only then this would be a three-arm study, which this is not. Please review the entire paper and edit accordingly. Second, please be consistent in your use of ‘DPV VR’ versus ‘DPV ring’ throughout the paper.

Sincerely,

Academic Editor